# Targeting Epithelial Mesenchymal Plasticity in Pancreatic Cancer: A Compendium of Preclinical Discovery in a Heterogeneous Disease

**DOI:** 10.3390/cancers11111745

**Published:** 2019-11-07

**Authors:** James H. Monkman, Erik W. Thompson, Shivashankar H. Nagaraj

**Affiliations:** 1Institute of Health and Biomedical Innovation, Queensland University of Technology, Brisbane, QLD 4059, Australia; e2.thompson@qut.edu.au; 2School of Biomedical Sciences, Queensland University of Technology, Brisbane, QLD 4059, Australia; 3Translational Research Institute, Brisbane, QLD 4102, Australia

**Keywords:** pancreatic cancer, epithelial mesenchymal plasticity, target discovery, review

## Abstract

Pancreatic Ductal Adenocarcinoma (PDAC) is a particularly insidious and aggressive disease that causes significant mortality worldwide. The direct correlation between PDAC incidence, disease progression, and mortality highlights the critical need to understand the mechanisms by which PDAC cells rapidly progress to drive metastatic disease in order to identify actionable vulnerabilities. One such proposed vulnerability is epithelial mesenchymal plasticity (EMP), a process whereby neoplastic epithelial cells delaminate from their neighbours, either collectively or individually, allowing for their subsequent invasion into host tissue. This disruption of tissue homeostasis, particularly in PDAC, further promotes cellular transformation by inducing inflammatory interactions with the stromal compartment, which in turn contributes to intratumoural heterogeneity. This review describes the role of EMP in PDAC, and the preclinical target discovery that has been conducted to identify the molecular regulators and effectors of this EMP program. While inhibition of individual targets may provide therapeutic insights, a single ‘master-key’ remains elusive, making their collective interactions of greater importance in controlling the behaviours’ of heterogeneous tumour cell populations. Much work has been undertaken to understand key transcriptional programs that drive EMP in certain contexts, however, a collaborative appreciation for the subtle, context-dependent programs governing EMP regulation is needed in order to design therapeutic strategies to curb PDAC mortality.

## 1. Pancreatic Cancer, Tumour Heterogeneity, and Carcinoma Vulnerabilities

Pancreatic cancer (PC) is the fourth most common cause of cancer-related deaths in Western societies, with 57,000 new cases annually, resulting in nearly 46,000 deaths in North America alone [1]. The most common type of PC is Pancreatic Ductal Adenocarcinoma (PDAC), which arises in the ductal epithelium of the exocrine tissue responsible for secreting pancreatic digestive juices. Late detection combined with early metastatic spread have limited gains in overall survival relative to other cancers such that PDAC mortality has the potential to surpass that of both colorectal and breast cancers by 2030 [2]. PDAC research therefore aims to define better diagnostic markers and novel therapeutic avenues, however is significantly complicated by the clinical heterogeneity present both within and between patient tumours. This emphasises the need for more integrative approaches aimed at developing a better understanding of targetable processes in PDAC tumourigenesis.

Cancer is a genetic disease caused by the accumulation of somatic mutations, resulting in a functional imbalance between tumour suppressive and oncogenic signals [3]. While transformed cells retain characteristics of the host to efficiently avoid being detected as foreign by the immune system, many aberrant phenotypes caused by genetic mutations and dysregulated signaling potentially render these cells susceptible to selective therapeutic interventions. Extensive examinations of the molecular traits of PDAC aimed at identifying such vulnerabilities have been conducted to date. Indeed, genomic and transcriptional profiling of patient tumours as part of large-scale studies by the The Cancer Genome Atlas (TCGA) and International Cancer Genome Consortium (ICGC) have allowed for insights into the scale of inter-tumour heterogeneity in a breadth of patient cohorts [4,5,6].

These studies have identified four major genetic aberrations common to pancreatic tumours [7,8,9]. 90% of tumours carry gain-of-function mutations in KRAS2, activating proliferative and cell survival pathways, whilst 95% contain either partial or complete inactivating mutations in CDKN2A, contributing to loss of cell cycle regulation, furthering proliferation. TP53, responsible for responding to DNA damage and inducing apoptosis, is altered in 60% of cases. SMAD4 inactivation is also common in pancreatic cancer development, and is found in 50% of patient cancers, disrupting the tumour suppressive signals of TGF*β*, aiding proliferation [10]. As well as these four common driver mutations, genomic sequencing of tumours has identified an additional panel of consistently mutated genes [6]. These genetic mutations implicate pathways often dysregulated in cancer, including KRAS, TGF*β*, WNT, NOTCH, ROBO/SLT, G1/S, SWI-SNF, and chromatin/DNA/RNA modification and repair.

Transcriptional profiling of PDAC tumours has allowed researchers to define discrete regulatory mechanisms within these networks that are associated with particular prognostic indices in different molecular subtypes of PDAC, which include squamous, pancreatic progenitor, immunogenic and aberrantly differentiated endocrine/ exocrine tumours [6]. Such classification schemes may provide clinical value by aiding in patient treatment regimen selection and planning [11], however, to date they have provided limited clinical value due to lack of targetable phenomena. It is important to note that while these studies have aimed to characterise changes within carcinoma cells, the excessive presence of desmoplastic stroma may confound these results. Indeed, microdissection of the tumour from its associated stroma has allowed the retrospective re-evaluation of large-scale transcriptional profiling efforts, highlighting the overwhelming contribution of stromal contamination to many such studies. Deconvolution based on laser capture microdissection and RNASeq profiling of 60 matched tumour/stroma pairs suggested that ICGC and TCGA samples contained stromal fractions of 46% and 55%, respectively, highlighting difficulties in deriving definitive conclusions from whole tumour analyses [12].

Such studies are invaluable as a means of understanding the intertumoural heterogeneity that exists between patients, and they form a strong set of public data that have been analysed to better appreciate the diversity of tumour presentation [13]. An increasing focus on single cell analytic technologies has yielded exciting opportunities to understand the contributions that individual cells make towards intratumoural heterogeneity, tumour progression, and patient outcomes [14,15]. These studies highlight the need for efforts aimed at distinguishing the heterogeneous nature of a tumour’s biology from that of the surrounding host tissue in which it propagates, so as to be better able to exploit cancer specific vulnerabilities [16].

As such, it is not surprising that the interactions between neoplastic epithelial cells and host myofibroblast and stellate populations, which can promote stromal inflammation, are increasingly being recognised. This desmoplastic reaction, which accounts for up to 90% of PDAC tumour volume, has pro-tumourigenic properties by leading to increased tissue stiffness and hypoxia as well as by providing physical barriers to both immune surveillance and chemotherapeutic penetrance [17,18,19]. The fibrillar collagen, hyaluronic acid and fibronectin rich extracellular matrix (ECM) deposited by stromal cells contains many soluble cytokines and growth factors secreted by both cancer and stromal compartments and contributes to both tumour initiation and progression [20,21,22,23]. Resident cells are forced to interact within this dynamic tumour microenvironment and are subject to stimuli that influence cell phenotypes in both stromal and carcinoma components. Such stimuli may propagate the invasion and dissemination of carcinoma cells by inducing epithelial mesenchymal plasticity (EMP), and thus this process is considered an important vulnerability that, when effectively targeted, may curb tumour progression [24,25].

## 2. EMP and PDAC Progression

EMP is often separated into two distinct but related processes—the forward process of epithelial-mesenchymal transition (EMT), and the reverse process of mesenchymal-epithelial transition (MET) [26]. These programs serve to describe the plasticity within epithelial cells that enables them to dedifferentiate into a more motile mesenchymal state, thereby allowing them to more effectively migrate. EMP is thought to play a significant role in several stages of tumour formation [27] and progression [28]. Initially, this plasticity allows tumour cells to detach and migrate from their site of origin (invasion), gaining access to lymphatic and blood vessels (intravasation), and then penetrating distant sites (extravasation), to form metastases.

A litany of reviews regarding different facets of EMP in PDAC, have been written, including those focused on molecular mechanisms of EMP regulation and metastasis [29,30,31,32,33,34,35,36], the role of epigenetic regulation [37], therapy development and resistance [38,39,40,41,42], microRNA regulation [43,44], and cancer stem cell generation [45,46,47,48,49]. This review thus focuses on some of the ongoing controversy surrounding in vivo evidence of EMP and the limitations of current approaches, highlighting the need to integrate a greater diversity of published EMP molecular regulators.

Development of PDAC frequently progresses undetected, remaining asymptomatic until it becomes an advanced stage of disease. Non-invasive precursor lesions formed either by epithelial proliferations or mucinous cysts in the pancreatic ducts, termed pancreatic intraepithelial neoplasia (PanINs), or intraductal papillary mucinous neoplasms (IPMNs), respectively, mark the onset of a histologically definable neoplasm in PDAC [50]. Such neoplasms, namely PanINs, progress through stages of dysplasia within the ductal epithelium, giving rise to the most common form of PDAC, pancreatic ductal adenocarcinoma (PDAC). The full breadth of factors that contribute to the invasive and metastatic behaviour of PDAC are vast. In this form of PDAC, there is very little latency between primary tumour formation and local and distant metastasis, implying that PDAC carcinoma cells may be readily equipped to invade and disseminate from a very early stage of development [51,52].

Invasive regions of human carcinomas are typically characterised by the presence of tumour-derived, fibroblast-like cells expressing mesenchymal markers such as vimentin, fibronectin and N-cadherin, with decreased expression of epithelial adhesion molecule E-Cadherin and increased nuclear beta-catenin relative to surrounding cells [53,54,55,56,57]. Decreased expression of E-cadherin has been shown to correlate with invasive and undifferentiated PDAC [58]. Furthermore, PDAC patients with tumour cells that express decreased E-cadherin and higher amounts of vimentin, s100A4, fibronectin and SNAI1 are more likely to have distant metastases, lymph node invasion and lower overall survival [54,59,60,61,62]. The EMP inducing transcription factor (TF) *TWIST1* has been shown to be upregulated in PDAC compared to match normal tissues [63], and *SNAI1* mRNA levels in PDAC fine needle aspirates are significantly correlated with lymph node and perineural invasion as well as with poorer survival [64]. A mediator of transforming growth factor beta (TGF*β*) signaling, SMAD3, was also shown to accumulate in the nucleus of PDAC samples, and was correlated with higher grade tumours and lymph node metastasis, indicating a role for TGF*β* in driving EMP in vivo [65]. Solitary infiltrating cancer cells displaying low E-cadherin and increased vimentin expression have proven to be significant prognostic indicators in resected clinical specimens from PDAC patients [66]. Tumour budding cells in PDAC have been observed with increased levels of *ZEB1* and *ZEB2*, and reduced levels of E-cadherin and *β*-catenin, indicative of EMP mediated local invasion. *ZEB2* overexpression in tumour-stroma associated cells also correlated with pathological assessment of tumour size, and lymph node metastasis [67]. Such striking pathology provides some of the clearest evidence for the role of EMP in PDAC progression.

While this clinical evidence strongly supports a role for EMP in mediating cancer invasion, the inability to accurately follow carcinoma epithelial dedifferentiation in vivo has led to some debate surrounding the extent of its role in tumour progression [68,69]. Such debate has necessitated the use of genetically engineered mouse models (GEMMs) to trace the role of EMP in cancer progression, specifically the pancreatic epithelium conditional *Kras*/*P53* mutant (PKCY) mice Lineage labelling of epithelial cells in this spontaneous PDAC model has allowed researchers to track these cells as they adopt mesenchymal properties and migrate away from the primary tumour into the circulation, seeding liver metastases [70]. In one study, EMP was detected in 42% of labelled PDAC epithelial cells, as assessed by the expression of EMP markers Zeb1 or Fsp1 and/or lack of E-cadherin. These cells were mostly observed in regions of inflammation, supporting the idea that EMP is driven by inflammatory interactions within the tissue microenvironment. Interestingly, some labelled epithelial cells that had undergone EMP displayed evidence of delamination and fibroblast morphology prior to tumour formation, and were otherwise indistinguishable from host stromal cells [70]. This is supportive of the very early, integral role that EMP may play in PanIN formation prior to tumour development.

Further studies in this same PDAC mouse model have shown that suppression of EMP via the knock-out of *Twist1* or *Snai1* TFs does not reduce metastasis, despite the decreased expression of EMP markers and increased cell proliferation as evidence for EMP ablation [71]. Equivalent numbers of lineage labelled epithelial cells were found in circulation and in metastases regardless of *Twist*/ *Snai1* knockout, suggesting that other mechanisms are involved in PDAC cellular invasion. PDAC cells do not possess a strong epithelial phenotype however, and may thus be insensitive to the loss of *Snail* TFs, which are potent repressors of epithelial programs but are less efficient in inducing mesenchymal properties. This possibly explains why *Snail* is dispensable for EMP and metastatic progression in this model [71,72], and points towards alternative mechanisms of EMP induction that may be driving factors in this PDAC system.

Indeed, there is evidence that the *Zeb1* TF is largely responsible for driving EMP in this GEMM model of PDAC development [73]. *Zeb1* ablation in PDAC cells was not found to affect *Twist1* expression, however it was associated with decreased *Zeb2*, *Slug* and a slight reduction in *Snai1* expression. *Zeb1* depleted tumours were better differentiated, indicating less local invasion, and showed significantly reduced metastasis when compared to control PDAC mice [73]. This is in direct contrast to depletion of *Twist1* or *Snai1*, which did not affect metastasis in this model system, highlighting the importance of recognising the context and tissue specific drivers of EMP.

Subsequent investigations aimed at overcoming the limitations of identifying single EMP regulatory TFs has shown that lineage labelled cancer cells are able to metastasize without expression of *α*Sma or Fsp1, both of which are thought to be robust markers of EMP activation in this model [74]. Indeed, larger metastatic nodules were found containing exclusively cells that had never expressed αSma or Fsp1, while micrometastatic clusters of 3–5 cells were shown to have undergone EMP. Such evidence, combined with the fact that *Zeb1* depletion in previous studies resulted in only a 50% reduction in metastasis underscores the pitfalls of seeking to identify individual master regulators and markers of such a complex process. Adding to this complexity, the emerging importance of hybrid EMP phenotypes, in which the expression of both epithelial and mesenchymal markers may occur at levels that are insufficient to drive the reporter constructs used in such lineage tracing models, adds a further technical challenge [75,76,77].

More recent attempts to understand EMP in individual PDAC cells has shown the activation of EMP transcriptional programs within certain subsets of tumour cell populations [14]. This study highlighted a clear role for cytokines from the stromal compartment in inducing EMP in certain PDAC cell lines, and indicated that EMP activation could be observed in discrete tumour gland subunits with prognostic utility. These models have provided considerable insights into the diverse mechanisms of PDAC development, and highlight that there are context-dependent EMP programs involved in both local invasion and metastatic dissemination that require further examination [72,78].

## 3. In Vitro EMP Models and Exogenous Stimuli

While GEMMS, in particular the PKCY model of spontaneous PDAC formation, are currently the gold standard for studies of the biology of EMP in tumourigenesis, *in vitro* studies form the basis for the majority of our current molecular understanding of intracellular events which occur in EMP. Many publicly available and in-house generated cell lines are used to study PDAC, but only a very limited number of these undergo well-characterised, stimulus-driven transitions that mimic the pathophysiological induction of EMP. This is perhaps consistent with the limited number of EMP events witnessed in in vivo models, highlighting the difficulties of studying such a dynamic process.

EMP is modulated by TGF*β*, receptor tyrosine kinases (RTK) ligands, WNT ligands, interleukins, hypoxia via HIF1*α* signaling, as well as HIPPO, NOTCH signaling. Their mechanisms and specific impact on downstream EMP targets have been comprehensively reviewed elsewhere, however our understanding of their subtleties is on-going [79,80]. TGF*β* acts as a tumour suppressor in normal tissue and early stage disease by regulating cell proliferation and inducing apoptosis through canonical signaling pathways, however this activity is lost as cellular transformation progresses [81,82,83,84,85]. Indeed, TGF*β* is a potent activator of EMP in PDAC cells when its tumour suppressive signals are disrupted through SMAD4 mutations, found in 50% of PDAC tumours [81,86]. Similarly, activating *KRAS* mutations found almost ubiquitously in PDAC cooperate with TGF*β* signaling to hyperactivate downstream RAS/RAF MAPK pathways to induce EMP [87]. While TGFB activates the greatest number of EMP signaling pathways, and may thus be considered a major driver in PDAC, the activation of additional pathways shown in Figure 1 by RTK, WNT and interleukin ligands may provide additional layers of crosstalk. Activation of SMAD, MAPK, PI3K, STAT, and NFκB pathways are commonly demonstrated in PDAC EMP research, however the relative extent to which each pathway governs EMP is unclear, as many studies evaluate these pathways independently [29,88,89,90,91,92,93,94].

These complex pathways ultimately serve to influence transcriptional programs that co-operate directly and indirectly to control the plasticity that exists between epithelial and mesenchymal phenotypes of carcinoma cells (Figure 1). Of note is the increasing recognition for the role of long non-coding RNAs (LncRNA) and micro-RNAs (miRNA) in EMP regulation. Among the cells that do undergo EMP-like transitions, there is a degree of selectivity for the ligands that are able to activate these EMP programs, and this is reflected in the limited number of commercial cell lines that are commonly manipulated within the field. This is consistent with the level of heterogeneity reported in PDAC, and suggests discrete differences in steady state signaling, which may predispose a given cell’s response or resistance to exogenous stimuli.

EMP is induced by stimuli shown within arrows on the left in order of potency. These signals activate signal transduction pathways that cooperate directly and indirectly to translocate signals to the nucleus (braced) to regulate EMP transcription factors, long non-coding RNAs (LncRNA), and micro RNAs (miRNAs).These factors then modulate EMP by discrete regulation of epithelial (Red Box) and mesenchymal (Green box) cellular properties, which in turn influence migration and invasion. Transforming growth factor (TGFB) activates the greatest number of these pathways, including direct cytoskeletal regulation by RhoA, aswell as canonical SMAD and non-canonical p38/JNK, MEK/ERK MAPK pathways and PI3K/AKT. Receptor tyrosine kinase (RTK) signaling is induced by binding of growth factor (GF) ligands such as EGF, IGF, FGF, HGF or VEGF and activates RAS/MEK/ERK, PI3K/AKT/NFκB and downstream SRC pathways. WNT signaling also modulates EMP by downstream stabilisation of B-catenin and subsequent nuclear translocation for EMP program activation by TCF/LEF transcription factors. Interleukins (ILs) can also induce EMP programs via STAT3 signaling. Additional mediators of EMP include Hypoxia, Hedgehog, Notch and Integrin signaling (not shown), and highlight the context dependent activation of EMP from micro-environmental cues.

While most studies rely upon knockdown and over-expression approaches to demonstrate the function of proteins in the context of cell migration, proliferation and EMP transitions, relatively few studies have investigated these targets in the context of the physiological induction of EMP in response to exogenous stimuli. Among PDAC cell lines, L3.6pl cells have been shown to respond to VEGF treatment [95], while the inflammatory cytokines TNF-α and IL-1*β* drive EMP in PaTu 8988T and AsPC-1 cells via Hedgehog signaling [96]. Collagen 1 also stimulated L3.6pl and BxPC-3 cells to become more invasive through interaction with DDR1 [97], and BMP2 was able to elicit a similar response in BxPC-3 cells [98]. PANC-1 cells are a well characterised model of inducible EMP, first shown by Ellenrieder et al to undergo a bidirectional change in response to TGF*β* alongside CAPAN-1, COLO-357, IMIM-PC1 [99], HPAF-II, and CAPAN-2 cells [100]. PANC-1 cells have since been repeatedly modelled with regard to their EMP response, which has been shown to be inducible in response to TGF*β*, TNF-*α*, HGF, or hypoxia through differing mechanisms [101,102,103,104]. *SNAI1* appears to be a major driver in this model, being heavily regulated at the transcript and protein level, despite modest changes in E-cadherin and Vimentin proteins [105]. EMP is thus invariably the result of exogenous stimuli that activate discrete but conserved cellular pathways through novel intermediates that are an ongoing focus of basic cancer cell biology research.

## 4. Pre-Clinical Discovery of EMP Targets

As a result of the complexities of discerning cancer biology from native processes in vivo, the use of cell lines derived from primary tumours are a valuable means of modelling the molecular and phenotypic properties of cancers. Extensive investigation has been performed using gene silencing and overexpression approaches to evaluate the role that particular molecules have in regulating or effecting the EMP phenotypes of PDAC cells, however a concise summary of novel targets in the PDAC EMP field has to date been lacking. Thus, this review provides an exhaustive overview of such research as a platform for their integration, and progressive evaluation. The function of these candidate molecules can be broadly separated into secreted/soluble products (Table 1), receptors (Table 2), other membrane associated proteins (Table 3), cytoskeletal adaptors (Table 4), kinases (Table 5), intracellular mediators (Table 6), transcription factors (Table 7) and post transcriptional controllers (Table 8). The candidates shown were selected by searching Pubmed for the terms ‘pancreatic’ and ‘epithelial’, and articles investigating a novel candidate’s impact on EMP phenotypes were manually curated. These effectors have been characterised to varying extents for their influence on invasion, migration, xenograft tumour growth, prognostic associations, and impact on known EMP signaling pathways. The proposed mechanisms of candidates and assays used to assess such effects are shown within tables and may be used to gauge where further support may be warranted to confirm and extend such findings. Due to the inherent variation in models used, the statistical power granted by IHC for varying sized patient cohorts with accompanying clinical information, and the level of EMP as a primary context, it is difficult to draw direct conclusions regarding pivotal significance within the field and clinical importance from such singular studies. Candidate expression in primary patient material that correlated with lymph-node metastasis are shown in bold within tables, and provide the best surrogate for their role in EMP mediated invasion, and include membrane bound proteins IGFBP2, ITGB4, CEACAM6 [106,107,108]. The use of IHC to capture dynamic EMP processes may be limited however, as shown in the case of LIN28B, where its expression is both induced by TGFβ and high in PDAC tissue, despite its role to suppress the pro-EMP non-coding RNA LET7a [109,110]. Such studies highlight both the utility and limitations of the links between in vitro assays and clinical material, and emphasise the need for both wider cohorts of patient material for validation and the development of GEMM models to strengthen findings in a standardized manner.

Figure 2 illustrates the proposed activity of some of these novel candidates, and how they may positively or negatively regulate discrete EMP signaling pathways. Of note are several candidates that converge to positively regulate EMP migratory phenotypes through FAK/Src and FAK/PI3K signaling, including the 5HT receptor and mucins, as well as EEF2K, USP22, and ZIP4. Their complete mechanisms of action and prevalence in PDAC tissue remain to be elucidated, however their inhibition may curb carcinoma invasion by blocking FAK activation and subsequent EMP modulation. Similarly, candidates participating in stability of EMP signaling and TF activity provide targets to modulate the EMP process specific for carcinoma cells. AURKA kinase has been shown to participate in a positive feedback loop with stabilization and activity of TWIST1, while PEAK1 and NES have been implicated in stabilization YAP/TAZ and SMAD TF activity. The discovery of discrete EMP regulation and development of combinatorial inhibitors may provide the opportunity for more personalized therapeutic approaches to curb metastatic disease.

EMP and cell migration (GREEN boxes) is induced through cell surface proteins (ITG, 5HTR, MUC, BLT2, SEMA3C, RTK, TGF*β*R) (RED) to activate signaling pathways (ORANGE boxes, blue arrows). These pathways are influenced by novel mediators (BLUE boxes) through activation (GREEN arrows) or inhibition (RED T) of known signaling members, however complete mechanisms of action remain to be elucidated. For full details, evidence of proposed mechanism and references of novel mediators, see tables below. Note signaling pathways shown have had intermediates removed for ease of visualisation.

## 5. Conclusions

Overall, investigation of the fundamental biology of EMP aims to combat local and metastatic invasion by providing a better understanding of the processes that allow cancer cells to dissociate from their epithelial adhesions to spread. EMP is a prominent driver of PDAC progression, thus highlighting the importance of our understanding of the subtleties of its regulation. The ability of EMP programs to direct cancer cells towards a drug resistant and migratory lineage capable of seeding local and distant recurrence presents a significant barrier to current treatment regimens. Therefore, the identification of new candidate molecules regulating these processes are crucial to inform targeted therapies and provide insights into the vulnerabilities of heterogeneous populations of tumour cells present in PDAC.

It is clear from this ever-growing list of EMP effectors in PDAC cells alone, that much work remains to delineate their collective interactions within and beyond our current understanding on EMP signaling pathways. While candidates have been shown to play roles in aspects of EMP signaling and associated phenotypes, significant support is required for their mechanisms of action to make concrete conclusions about their directive actions in cancer. Our understanding of receptor mediated canonical signaling through PI3K/AKT, MAPK, NFκB and other well studied cell cycle pathways has required decades to tease apart, and the subtleties of EMP programs provides a similar challenge. Open source integrative tools such as Reactome [200], WikiPathways [201], String [202], and Cytoscape [203] provide platforms for researchers to combine such analyses to build upon our current understanding and fill knowledge gaps in the field of cancer biology. In this way, progress may be made to better understand and discover properties that may be modulated in concert to control EMP in cancer.

*In vitro* and xenograft tumour modelling and manipulation of target molecules often demonstrates a role in cancer cell migration and tumour formation, however stronger evidence for their physiological role in regulating EMP, metastasis and therapy resistance may require GEMMs. The use of *in vivo* manipulation of PDAC GEMM models using targeted CRISPR approaches may be such a route towards a system that better recapitulates the spontaneity and heterogeneity of human tumours [204].

## Figures and Tables

**Figure 1 cancers-11-01745-f001:**
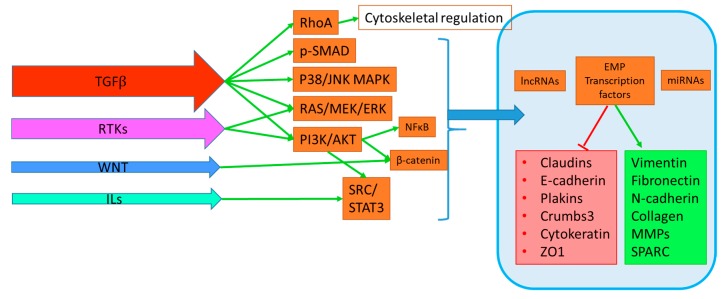
Simplified overview of cooperating signaling pathways in EMP.

**Figure 2 cancers-11-01745-f002:**
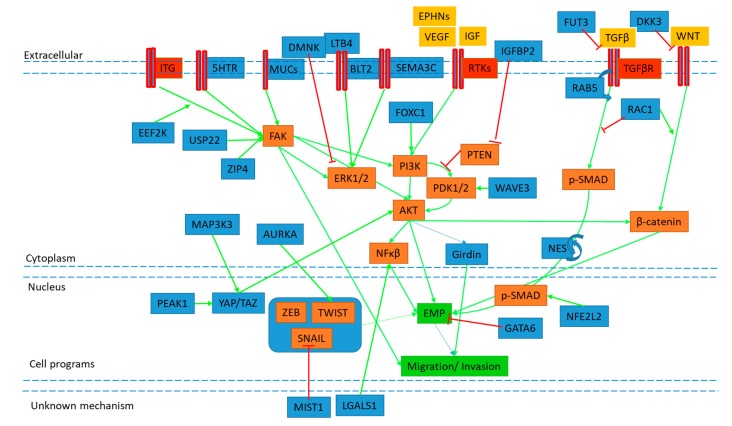
Simplified overview of the proposed mechanisms of novel candidates.

**Table 1 cancers-11-01745-t001:** Soluble and secreted factors that influence EMP. This table describes novel candidates that may be secreted within the ECM and act either directly through ligand-receptor interactions, or through mechanisms that remain to be demonstrated. Candidates that exhibit clinical correlation with lymph node metastasis are shown in bold.

Cell Line	Target	EMT Regulation (Direct or Indirect Observation)	KD/KO/Over-expression	Pathway/Mechanism	Functional Assay	Human Prognostic Association	EMT Activator	Reference
BxPC-3	DKK3	Negative, Direct	Over-expression	DKK3 is overexpressed in tumour and is antagonist of WNT ligand activity, preventing nuclear translocation of *β*-catenin and EMP under hypoxia	Transwell assays, chemo-resistance, IHC in 75 matched PDAC v normal samples, xenograft growth	Not performed	Hypoxia	[111]
**ASPC-1, PANC-1**	**IGFBP2**	**Positive, Direct**	**siRNA/Over-expression**	**IGFBP2 activated NF-κB through PI3K/AKT/IKK, inhibited by PTEN**	**WB, Transwell assays, orthotopic growth, IHC in 80 patient PDAC and lymph node samples**	**Survival and lymph node metastasis**	**-**	**[106]**
PANC-1	LTB4	Positive, Direct	siRNA	LTB4 induced EMT through receptor BLT2 and ERK1/2 activation	WB, Transwell assays	Not performed	LTB4	[112]
Patu8988, PANC-1	DMKN	Positive, Indirect	shRNA	Knockdown reduced p-STAT3 and EMT increased ERK1/2, AKT	Proliferation, Transwell assays, Xenograft, IHC in 44 patient PDAC tumours	Correlated with T stage	-	[113]
PANC-1	LGALS1	Positive, Direct	shRNA/Over-expression	LGALS1 IHC expression correlated with MMP9 and Vimentin in PDAC. PSC LGALS1 promoted cancer cell EMT and activation of NF-κB	Xenograft, Proliferation, Invasion, IHC in 66 PDAC tumours	Not performed		[114]
BxPC-3, CFPAC	SEMA3C	Positive, Indirect	shRNA/Over-expression	SEMA3C knockdown suppressed EMT and tumourigenesis, and activation of ERK1/2 signaling	Proliferation, migration, Scratch wound, Xenograft, IHC in 118 PDAC tumours	Stage, survival, recurrence		[115]
Capan-1	FUT3	Positive, Direct	shRNA/Over-expression	FUT3 knockdown impeded proliferation, migration, tumour growth and TGF*β* induced EMT	Proliferation, Scratch wound, Transwell assays, Xenograft	Not performed	TGF*β*	[116]
**PANC-1, MIAPaCa-2, Capan-2**	**MIF/ NR3C2**	**Positive, Indirect**	**siRNA/Over-expression**	**MIF induces miR-301b, targeting NR3C2, inducing EMT and chemo sensitivity**	**PDAC transcriptome by array, IHC of 173 PDAC, Proliferation, Colony formation, Transwell assays, chemo-resistance**	**NR3C2 inversely prognostic by RNA and IHC**	**-**	**[117]**
PANC-1, BxPC-3	WNT5A	Positive, Direct	siRNA, Over-expression	Wnt5a expression induced EMT and invasion and was elevated in PDAC by IHC	Scratch wound, Transwell assays, WB, orthotopic growth, IHC of 134 PDAC v normal	No	-	[118]
PANC-1	LCN2	Negative, Indirect	Over-expression	LCN2 expression correlated with better survival and lower EMT state	IHC of 60 PDAC tumours, Transwell assays	Protective by IHC	-	[119]
MIAPaca-2, BxPC-3, SUIT-2	NOV	Positive, Indirect	shRNA/Over-expression	NOV expression high in PDAC by IHC, and induced EMT phenotypes in vitro/in vivo	Colony formation, soft agar, Proliferation, Transwell assays, in vivo metastasis	Not performed	-	[120]
PANC-1, BxPC-3	CCL18	Positive, Direct		CCL18 expressed in mesenchymal and cancer cells, and induced EMT	WB, Transwell assays, IHC of 62 PDAC tumours, serum ELISA from PDAC patients	Survival	-	[121]
**PANC-1, BxPC-3**	**TUFT1**	**Positive, Indirect**	**siRNA/Over-expression**	**TUFT1 expression correlated with T stage and lymph node metastasis by IHC, RNA expression correlated with HIF1a, SNAI1 and VIM**	**WB, Proliferation, scratch wound, Transwell assays, Xenograft, IHC of 63 PDAC tumours**	**Yes in TCGA by RNA**		**[122]**
SW1990, ASPC-1	OLR1	Positive, Direct	siRNA/Over-expression	OLR1 overexpressed in tumours and correlates with metastasis and poor survival, overexpression induced EMT	Transwell assays, scratch wound, Proliferation/apoptosis, IHC of 98 PDAC tumours	Yes survival by IHC and TCGA	-	[123]
MIAPaCa-2, PANC-1, ASPC-1, BxPC-3	LOXL2	Positive, Indirect	siRNA/Over-expression	LOXL2 IHC expression correlated with recurrence, depth of invasion and poor survival, and enhanced EMT in vitro	Transwell assays, IHC of 80 PDAC tumours	Yes by IHC	-	[124]
PANC-1, PK9	TFF1	Negative, Direct	siRNA	TFF only expressed in PanIN and intraductal neoplasia, not normal or invasive PDAC, knockdown activated EMT, loss of TFF in GEMM drove PanIN, PDAC and CAF infiltration	Transwell Invasion, Scratch wound, KC GEMM, IHC on small number of samples	Not performed	-	[125]

**Table 2 cancers-11-01745-t002:** Receptors. This table describes known receptors that may be activated to transduce signals required for EMP modulation. Candidates that exhibit clinical correlation with lymph node metastasis are shown in bold.

Cell Line	Target	EMT Regulation	KD/KO/Over-expression	Pathway/Mechanism	Functional Assay	Prognostic Association	EMT Activator	Reference
L3.6pl	VEGFR1 activation	Positive, Direct		RTK VEGFR-1 activation induced SNAI1/2, TWIST	E-cadherin/b-catenin localization/WB	Not performed	VEGF	[95]
PANC-1, MiaPaCa-2	HTR1B, HTR1D	Positive, Indirect	siRNA	5-HT receptor knockdown reduced uPAR and Src/FAK signaling and EMT	Scratch wound, Transwell, Colony formation	Not performed	-	[126]
PANC-1 HPAC	IGF1R	Positive, Indirect	siRNA	IGF1R overexpressed in PDAC by IHC, silencing inhibits AKT/PI3K, MAPK, JAK/STAT signaling pathways	Transwell assays, soft agar, Proliferation, apoptosis, IHC of TMA	Not performed	-	[127]
L3.6pl, BxPC-3	DDR1	Positive, Direct	siRNA/ Over-expression	DDR1 expression correlates with CHD2 expression by IHC, DDR1-b signals through SHC1 adapter to PYK2 to induce CDH2	Invasion, IHC of PDAC TMA	Not performed	COL1A	[97]
PANC-1	SMO	Positive, Indirect	siRNA	Hedgehog activated in tumourspheres, SMO knockdown inhibited CSC/EMT features properties	Proliferation, sphere formation, Transwell assays, Xenograft	Not performed	-	[128]
PANC-1, BxPC-3	EPHA4	Positive, Direct	siRNA	EPHA4 knockdown suppressed EMT, MMP2 activity	Gelatin zymography, Transwell assays, scratch wound, WB	Not performed	-	[129]
**CFPAC-1, AsPC-1**	**ITGB4**	**Positive, Direct**	**siRNA/Over-expression**	**ITGB4 IHC expression correlated with T stage, knockdown inhibited EMP**	**Transwell assays, WB, IHC of 134 PDAC tumours**	**Survival lymph node metastasis by IHC**	**TGF*β***	**[107]**
PANC-1, MiaPaCa2, Capan2	F2R	Positive, Indirect	shRNA	F2R (PAR1) expression associated with mesenchymal gene signature	Xenograft, Scratch wound	Not performed	-	[130]

**Table 3 cancers-11-01745-t003:** Membrane associated proteins. This table describes membrane bound proteins that may interact with other cells and the extracellular environment to sense cues that modulate EMP in a context dependent fashion. Candidates that exhibit clinical correlation with lymph node metastasis are shown in bold.

Cell Line	Target	EMT Regulation	KD/KO/Over-expression	Pathway/Mechanism	Functional Assay	Prognostic Association	EMT Activator	Reference
PANC-1	CDCP1	Positive, Indirect	siRNA	CDCP1 expression high in PDAC, induced by BMP4/ERK signaling, and knockdown inhibited EMT phenotypes	Scratch wound, Transwell, spheroid formation, chemo-resistance, IHC on 42 PDAC tumours	Not performed	-	[131]
Colo-357, Capan-1	MUC16	Positive, Indirect	siRNA, CRISPR/Cas9	MUC16 knockdown decreased FAK mediated AKT/ERK/MAPK activation, and EMT	Proliferation, migration, Colony formation, Xenograft	Not performed	-	[132]
MiaPaCa2	ANXA1	Positive, Indirect	CRISPR	ANXA1 KO downregulated miR196a, effected cell motility and liver metastases in vivo	Scratch wound, Transwell migration, Invasion, Xenograft	Not performed		[133,134]
**CFPAC-1, PANC-1**	**CEACAM6**	**Positive, Direct**	**shRNA, Over-expression**	**CEACAM6, regulated by miR-29a/b/c, required for EMT**	**Transwell assays, Xenograft, WB, IHC in 99 PDAC tumours**	**Lymph node metastasis**	**-**	**[108]**
SUIT-2, CAPAN-2	TM4SF1	Negative, Indirect	siRNA	TM4SF1 IHC expression protective, knockdown induced migration and decreased E-cadherin	Transwell assays, IHC in 74 PDAC tumours	Yes inversely prognostic by IHC	TGFβ	[135]
PANC-1, SW1990	DPP4	Positive, Indirect	siRNA/ Over-expression	DPP4 (CD26) knockdown suppressed EMT, *in vivo* growth	Proliferation, Transwell assays, Xenograft, WB	Not performed	-	[136]
PANC-1, AsPC-1	SLC39A4	Positive, Indirect	siRNA/ overexpression	SLC39A4 (ZIP4) IHC expression correlated with ZEB1 and EMT, increasing FAK and paxillin phosphorylation	Xenograft, Scratch wound, Transwell migration, Invasion, IHC of 72 paired PDAC v normal	Not performed	-	[137]

**Table 4 cancers-11-01745-t004:** Cytoskeletal adaptors. This table describes intracellular adapter proteins that may participate in and be required protein complex localization and transduction of signals that modulate EMP. Candidates that exhibit clinical correlation with lymph node metastasis are shown in bold.

Cell Line	Target	EMT Regulation	KD/KO/Over-expression	Pathway/Mechanism	Functional Assay	Prognostic Association	EMT Activator	Reference
**PANC-1, CFPAC-1**	**WASF3**	**Positive, Indirect**	**siRNA,**	**WASF3 (WAVE3) knockdown suppressed PDK2, downregulating PBK/AKT pathway and EMT**	**Proliferation, migration, Invasion, Scratch wound, IHC of 87 paired PDAC v normal**	**Lymph node metastasis**	**-**	**[138]**
PANC-1, AsPC-1, MiaPaCa-2	NES	Positive, Direct	shRNA/Over-expression	NES (Nestin) required for EMT and induced by TGF*β* in positive feedback loop promoting p-smad2	Xenograft, Transwell assays, IHC of GEMM	Not performed	TGF*β*	[139,140]
HPAF-II, PANC-04.03 PANC-1	DNM2	Positive, Indirect	siRNA, Over-expression	Upregulated by IHC in PDAC, DNM2/VAV1 interaction required for RAC-1 induced lamellipodia formation	Transwell assays, lamellipodia formation, xenograft, IHC of 85 PDAC tumours	Not performed	EGF (HPAF-II)	[141,142]
SUIT-2	RAB5A	Positive, Indirect	siRNA	RAB5 IHC expression correlated with invasion and CDH1, aids TGFβR endocytosis, stimulates FA turnover, prognostic in PDAC, breast, ovarian	Morphology, Proliferation, Transwell assays, IHC of 111 PDAC tumours	Survival IHC	-	[143]

**Table 5 cancers-11-01745-t005:** Kinases and Phosphatases. This table describes proteins with activity that may directly participate in signal transduction by phospho-regulation of intracellular substrates. Candidates that exhibit clinical correlation with lymph node metastasis are shown in bold.

Cell Line	Target	EMT Regulation	KD/KO/Over-expression	Pathway/Mechanism	Functional Assay	Prognostic Association	EMT Activator	Reference
PANC-1, MIAPaCa-2	EEF2K,	Positive, Direct	siRNA/Over-expression	EEF2K promotes EMT through TG2/*β*1 integrin/SRC/uPAR/MMP2 signaling	Scratch wound, Transwell assays, WB	Not performed	-	[144]
Patu8988, PANC-1, BxPC-3, Capan-1	CDK14	Positive, Direct	siRNA	Suppression of CDK14 reduced PI3K/AKT activation and EMT	Proliferation, Colony formation, Transwell assays	Not performed	-	[145]
HDPE	PRAG1	Positive, Indirect	siRNA/Over-expression	Phosphorylation of PRAG1 found in malignant cells, Over-expression induced JAK1/STAT3 mediated EMT	Transwell assays, phospho-WB	Not performed	-	[146]
BxPC-3, PANC-1	AURKA	Positive, Direct	shRNA	AURKA IHC expression high in PDAC, phosphorylates and stabilizes TWIST1 in positive feedback loop, promoting EMT	Sphere formation, migration, Proliferation, Xenograft, IHC on small PDAC cohort	Not performed	-	[63]
PANC-1, ASPC-1	MAP3K3	Positive, Direct	CRISPR	MAP3K3 (MEKK3) KO reduced EMT, CSC and migration, and YAP/TAZ transcriptional activity on AXL, DKK1, FosL1, CTGF	Transwell migration Invasion, Proliferation, Xenograft, ChIP	Not performed	-	[147]
PANC-1, COLO357	RAC1	Negative, Direct	siRNA/Over-expression	RAC1b inhibits canonical and non-canonical TGF*β* signaling, effecting MKK6-p38 and MEK-ERK-MAPK EMT activation	Migration, qPCR	Not performed	TGF*β*	[90,148]
HPAF-II, CAPAN-2	PTPN11	Positive, Direct	shRNA/Over-expression	PTPN11 (SHP2) activity enhances the effect of EGF on TGF*β* induced EMT, resulting in more complete EMT	Cell scatter, scratch wound, WB	Not performed	TGF*β*/EGF	[100]

**Table 6 cancers-11-01745-t006:** Enzymes and Co-factors. This table describes intracellular proteins that may directly or indirectly participate in pathways required for EMP modulation by other enzymatic control of substrate proteins. Candidates that exhibit clinical correlation with lymph node metastasis are shown in bold.

Cell Line	Target	EMT Regulation	KD/KO/Over-expression	Pathway/Mechanism	Functional Assay	Prognostic Association	EMT Activator	Reference
PANC-1	USP22	Positive, Direct	shRNA/Over-expression	USP22 expression correlated with Ezrin and FAK phosphorylation and EMT	Scratch wound, Transwell assays, WB	Not performed	-	[149]
779E, 1334 PDCL	EIF5A	Positive, Indirect	shRNA/Over-expression	Mutant KRAS induces EIF5A, stimulating PEAK1 mediated ECM signaling. PEAK1 binds YAP/TAZ driving stem TFs	Sphere formation, IP, WB	Not performed	-	[150]
AsPC-1, PANC-1	EIF4E	Negative, Indirect	siRNA	Knockdown of MNK effector, EIF4E, induced ZEB1 through repression of miR-200c, miR-141, MNK inhibitors induce MET	Collagen 3D, qPCR	Not performed	-	[94]
BxPC-3	RGCC	Positive, Direct	siRNA	RGCC regulated by HIF1α and required for hypoxia induced EMT	qPCR, WB	Not performed	hypoxia	[151]
PANC-1 MIA PaCa-2	SET	Positive, Direct	shRNA/ Over-expression	SET over-expression activated Rac1/JNK/c-Jun pathway and decreased PP2A activity, N-cadherin and EMT TFs up	Transwell assays, Colony formation, Xenograft tumour growth and liver metastases	Not performed	-	[152]
MiaPaCa2, SW1990, PANC-1, CFPAC1	GPX1	Negative, Direct	shRNA/Over-expression	GPX1 IHC expression lower in PDAC, silencing induced EMT and gemcitabine resistance through ROS activated PI3K/Akt/GSK3B/SNAIL, Over-expression sensitized in vivo	Transwell migration, chemo-resistance, Xenografts, IHC of 281 PDAC tumours, and 42 paired PDAC v normal	Yes inversely prognostic by IHC		[153]
BxPC-3, PANC-1, MiaPaCa2, PSN1	HDAC1	Positive, Indirect	siRNA	HDAC IHC expression and activity correlated with EMT phenotype	IHC, Transwell Invasion, IHC of 103 PDAC tumours	Survival by IHC	-	[154]
PANC-1 BxPC-3	Class I HDAC	Positive, Indirect	4SC-202 small inhibitor	HDACi (inhibition) blocked TGF*β* induced EMT in PANC-1, requiring BRD4 and MYC for effect of HDACi	Migration, sphere formation, Xenograft	Not performed	TGF*β* (PANC-1)	[155]
CFPAC-1, L3.7-2	PAFAH1B2	Positive, Direct	siRNA/Over-expression	PAFAH1B2 IHC expression higher in PDAC, HIF1a expression regulated PAFAH1B2 via direct promoter binding	Transwell migration, Invasion, orthotopic Xenograft/ liver metastases, HIF1a/PAFAH1B2 co-localization in PDAC, IHC of 124 PDAC tumours and 70 normal	Survival by IHC and TCGA	hypoxia	[156]
PANC-1, MIAPaCa-2	KDM4B	Positive, Direct	siRNA	KDM4B IHC expression correlated with ZEB1 in PDAC, knockdown inhibited TGFβ induced EMT in PANC-1 by regulating ZEB1 methylation	CHIP, scratch wound, Transwell assays, IHC of 49 PDAC tumours	Not performed	TGFβ	[157]
HPAC, BxPC-3, Colo357 PANC-1, MiaPaCa-2	SMURF2	Negative, Direct		SMURF negative regulator of TGF*β* induced EMT, suppressed by miR-15b	Scratch wound, Transwell assays, WB	Not performed	TGF*β*	[158]
CAPAN-1 PANC-1	CUL4B	Positive, Direct	miRNA	CUL4B IHC expression higher in PDAC, regulated by miR -300, required for Wnt/*β*-catenin induced EMP	qPCR, Transwell assays, Xenograft, IHC of 110 PDAC v normal	Not performed	-	[159]
PANC-1	KMT5C	Positive, Direct	siRNA	KMT5C (SUV420H2) expression higher in PanIN and PDAC, methylates H4K20me3,suppresses epithelial drivers FOXA1, OVOL2, OVOL2	Transwell assays, chemo-resistance, sphere formation,	Not performed	-	[160]
PANC-1	NOX4	Positive, Direct	siRNA	NOX4 IHC expression elevated in PDAC, aids ROS generation and TGF*β* induced EMT	Transwell assays, WB	Not performed	TGF*β*	[161]
BxPC-3	PAWR	Negative, Indirect	siRNA, Over-expression	PAWR (PAR4) suppressed in cisplatin resistant EMT cells, required PI3K/AKT signaling	Transwell assays, Proliferation, WB, Xenograft	Not performed	-	[162]
BxPC-3	PPM1H	Negative, Indirect	siRNA	PPM1H expression decreased by TGF*β*/BMP2 treatment, knockdown induced EMT	Proliferation, Transwell assays, WB, apoptosis	Not performed	TGF*β*, BMP2	[98]
PANC-1	HMGN5	Positive, Indirect	shRNA	HMNG5 silencing reduced Wnt expression	Xenograft, Transwell migration Invasion, WB,	Not performed	-	[163]
PANC-1	GOLM1	Positive, Direct	siRNA/ overexpression	GOLM1 (GP73) overexpression induced EMT and correlated with human metastasis and Xenograft growth	Xenograft, Transwell migration Invasion, Scratch wound, WB	Not performed	-	[164]

**Table 7 cancers-11-01745-t007:** Transcription Factors and Cofactors. This table describes transcription factors and cofactors that influence gene expression required for actions of EMP in their respective systems. Candidates that exhibit clinical correlation with lymph node metastasis are shown in bold.

Cell Line	Target	EMT Regulation	KD/KO/Over-expression	Pathway/Mechanism	Functional Assay	Prognostic Association	EMT Activator	Reference
PaTu 8988T, AsPC-1	GLI1	Positive, Direct	siRNA	GLI1 component of HH signaling, induced EMT by TNF-*α*/IL-1*β*, mediated through NF-κB pathway	Transwell assays, WB	Not performed	TNF-*α*/IL-1*β*	[96,165,166,167,168]
Colo-357, L3.7	FOXM1	Positive, Indirect	siRNA/Over-expression	FOXM1c activates uPAR promoter directly, inducing EMT	Scratch wound, Transwell migration, IHC of PDAC TMA v normal	Elevated in metastatic PDAC	-	[169]
BxPC-3, ASPC-1, PANC-1	TAZ	Negative, Direct	shRNA, Over-expression	TAZ required for EMT through TEA/ATTS TFs, activation correlates with suppression of NF2	Colony formation, Xenograft, Transwell assays, IHC of 57 PDAC v normal	Correlated with PDAC differentiation	-	[170]
PANC-1, CAPAN-1	YAP	Positive, Direct	shRNA/Over-expression	YAP expression associated with activation of AKT cascade and EMT	Transwell assays, chemo-resistance, WB	Not performed	-	[171]
PANC-1, BxPC-3	HSF1	Positive, Indirect	siRNA	p-HSF1 IHC elevated in PDAC, promotes invasion and is downregulated by p-AMPK	Transwell assays, scratch, WB, GEMM	Not performed	-	[172]
HPAC, MiaPaCa2	FOXC1	Positive, Indirect	siRNA/Over-expression	IGFR1 positively regulates FOXC1, activating PI3K/Akt/ERK, promoting migration, and EMT, and tumour growth	Xenograft, Transwell migration Invasion, soft agar	Not performed	IGF	[173]
PANC-1, SW1990	BHLHA15, Direct	Negative, Direct	Over-expression	BHLHA15 (MIST1) Over-expression suppressed tumour growth & metastases. Caused MET by suppressing SNAIL indirectly	Transwell migration, Invasion, Xenograft and liver met	Not performed	-	[174]
PANC-1	KLF8, Indirect	Positive, Direct	siRNA, Over-expression	KLF8 IHC elevated in PDAC, directly induces FHL2 transcription via promoter binding	WB, Invasion	Not performed	-	[175]
GEMM	P73, Direct	Negative, Direct	GEMM	P73 deficiency led to stromal deposition and EMT in PDAC tumours, decreased BGN secretion, required for tumour suppressive functions of TGFβ	GEMM, Transwell assays	Not performed	-	[176]
GEMM	PRRX1	Positive, Direct	Overexpression	PRRX1 a/b have discrete functions in MET/EMT, knockdown suppresses tumour growth and EMT	GEMM tumour model, Xenograft	Not performed	-	[177]
**Capan-2**	**TRIM28**	**Positive, Indirect**	**Overexpression**	**TRIM28 Overexpression drove EMT and Invasion, correlated with T stage**	**Transwell assays, WB, Xenograft, IHC of 91 PDAC**	**Lymph node metastasis and survival**	**-**	**[178]**
PANC-1	ETS1	Positive, Direct	shRNA	ETS1 knockdown epithelialized PANC-1 cells	Scratch wound, adhesion, qPCR for EMT markers	Not performed	-	[179]
HDPE, COLO-357	NFE2L2	Positive, Direct	siRNA/Over-expression	NFE2L2 activation enhanced TGFβ induced EMT in both premalignant and malignant cells	Scratch wound, Transwell assays, WB, qPCR	Not performed	TGF*β*	[180]
PANC-1, HPAF-II	PDX1	Positive, Indirect	shRNA, GEMM	PDX1 has dual roles in premalignant and transformed cells. PDX1 expression is reduced in tumours and EMT	Colony formation, GEMMs, IHC of 183 PDAC	Inversely prognostic for survival	TGF*β* (PANC-1), HGF (HPAF-II)	[181]
PANC-1	BCL9L	Positive, Direct	siRNA/Over-expression	BCL9L knockdown prevented EMT and inhibited in vivo growth	Proliferation, Transwell assays, Xenograft	Not performed	TGF*β*	[182]
GEMM	ETV1	Positive, Direct	Overexpression	ETV1 induces SPARC, required for tumour growth and metastasis in vivo, EMT *in vitro*	Xenograft, Invasion	Not performed	-	[183]
**ASPC-1, SW1990**	**EPAS1**	**Positive, Direct**	**siRNA**	**EPAS1 (HIF2α) IHC expression high in PDAC, and knockdown inhibited EMT**	**CHIP, Transwell assays, IHC of 70 PDAC**	**Lymph node metastasis, differentiation**	**-**	**[184]**
PANC-1 BxPC-3	SIX1	Positive, Indirect	siRNA/shRNA	SIX1 IHC expression elevated in PDAC, knockdown reduced migration and tumour size	Migration, EMT markers, PANC-1 Xenograft, CD44-/CD24+, IHC of 139 PDAC	No	-	[185]
Cfpac-1	GRHL2	Negative, Direct	siRNA	GRHL2 IHC expression elevated in normal duct and liver metastases, drives epithelial phenotype.	Proliferation, EMT markers, Colony and sphere formation, drug resistance, IHC of 155 PDAC	No	-	[186]
PaTu8988S	GATA6	Negative, Direct	shRNA/Over-expression	GATA6 IHC expression low in PDAC, Silencing induced EMT	chemo-resistance, IF, Invasion, Xenograft, IHC of 58 PDAC	Inversely prognostic for survival	-	[187]

**Table 8 cancers-11-01745-t008:** Post transcriptional effectors. This table describes factors that may post transcriptionally modulate EMP by controlling stability of mRNA and hence expression of effector proteins. Candidates that exhibit clinical correlation with lymph node metastasis are shown in bold.

Cell Line	Target	EMT Regulation	KD/KO/Over-expression	Pathway/Mechanism	Functional Assay	Prognostic Association	EMT Activator	Reference
Miapaca-2, PANC-1, Patu-8988	HNRNPA2B1	Positive, Direct	shRNA/Over-expression	Knockdown epithelialized cells, Over-expression drove EMT through ERK/SNAI1 pathway	Cell viability, Transwell assays, PANC-1 Xenograft, EMT markers	Not performed	-	[188]
SW1990, BxPC-3	YTHDF2	Negative, Direct	shRNA	Knockdown reduced p-AKT, p-GSK-3b, promoted EMT, YAP knockdown reversed effect	Proliferation, Colony formation, Invasion, adhesion	Not performed	-	[189]
Panc-1, Patu8988	Lnc TUG1	Positive, Direct	shRNA	Lnc TUG1 sponges miR-382, preventing repression of ezh2	Colony formation, Transwell assays, WB	Not performed	-	[190]
Gemcitabine resistant BxPC-3	DYNC2H1-4	Positive, Direct	siRNA	Lnc DYNC2H1-4 sponges miR-145, upregulating ZEB1, MMP3 and other CSC markers	Transwell assays, CSC markers, Xenograft	Not performed	-	[191]
ASPC-1, BxPC-3, PANC-1	miR-23	Positive, Direct	miRNAs	miR -23 promotes EMT by regulating ESRP1, miR-23 required for TGFβ induced EMT	WB, Transwell assays, Xenograft, qPCR of 52 paired PDAC tumour v normal	Survival by RNA	TGFβ	[192]
SW1990, PANC-1, BxPC-3, CAPAN-1	NORAD	Positive, Direct	shRNA/Over-expression	Lnc NORAD acts as ceRNA of miR-125a-3p, enhancing RHOa and EMT	Scratch wound, Transwell assays, Xenograft	Not performed	Hypoxia	[193]
Panc-1	Lnc H19	Positive, Direct	siRNA	H19 antagonised LET-7, inducing HMGA-2 mediated EMT	Transwell assays, scratch wound, WB	Not performed	-	[194]
ASPC-1, BxPC-3	LncRNA-ROR	Positive, Direct	shRNA/Over-expression	LncRNA-ROR expression induces ZEB1 and EMT	Scratch wound, Transwell assays, Xenograft	Not performed	-	[195]
PANC-1, BxPC-3, COLO357	miR-100, miR -125b	Positive, Indirect	siRNA/CRISPR/Over-expression	TGFβ induced lnc-miR100HG, which codes for tumourigenic miR 100, miR125b and LET-7a. LIN28B also induced by TGFβ, suppresses LET-7a activity	miR Over-expression, Xenograft, Scratch wound, sphere formation, RNAseq, RIPseq	Survival by RNA	TGFβ	[109]
BxPC-3, PANC-1, CFPAC-1, SW1990	miR-361-3p	Positive, Direct	Over-expression	miR-361-3p downregulates DUSP2, preventing inactivation of ERK1/2	Orthotopic metastasis, Transwell assays	Survival by RNA	-	[196]
Sw1990	miR-1271	Negative, Direct	miR Mimics, Inhibitors	miR -1271 inhibited EMT and migration	Proliferation, Transwell migration invasion, xenograft	Not performed	-	[197]
Panc-1	LSM1	Positive, Indirect	Over-expression	Lsm1 (CaSm) induction induced EMT and proliferation, effecting apoptotic and metastasis gene expression	Proliferation, anoikis, Transwell assays, chemo-resistance, xenograft	Not performed	-	[198]
KPCY	MTDH	Positive, Indirect	siRNA	MTDH expression promoted CSC and metastasis, high cytoplasmic expression by IHC	Spheroid formation, orthotopic and metastatic xenograft models, IHC of 134 PDAC	Survival	-	[199]
**ASPC-1, HS766t, BxPC-3**	**LIN28B**	**Positive, Direct**	**shRNA**	**LIN28B IHC expression high in PDAC, suppression inhibited proliferation and EMT**	**Colony formation, Proliferation, migration, IHC of 185 PDAC tumours**	**Survival, stage, metastasis**	**-**	**[110]**

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
