# Peer review of "Targeting Epithelial Mesenchymal Plasticity in Pancreatic Cancer: A Compendium of Preclinical Discovery in a Heterogeneous Disease"

_cancers, 2019, doi:10.3390/cancers11111745_

Round 1
Reviewer 1 Report
The article by Monkman and colleagues reviews epithelial mesenchymal plasticity (EMP) in pancreatic cancer.
The topic is timely and of interest. There are, however, concerns with the present manuscript/ analysis.
The manuscripts reviews EMP in pancreatic cancer; the main part is an extensive overview of studies on pancreatic cancer cell lines that have analyses specific factors and their influence on EMP. This is obviously a large amount of work that has been done. What is somewhat missing is a ‘quality control’ of the included papers. How reliable were the effects on EMP analysed? Was EMP the major focus of the studies or only a ‘bystander’ result. Further, a graphical overview about the most common pathways analyses would be helpful; e.g. many analysed genes were related (directly or indirectly) to TGF-beta signalling. There are several incorrect statements, e.g. four major oncogenic pathways (actually, one oncogenic, three tumor suppressor); 95% inactivating mutations of CDKN2 (actually, mutated only partially and otherwise silenced). “a panel of 32 consistently mutated genes” is not correct. Probably it is meant “genes in 32 specific pathways”. “…PanINs, induce dysplasia within the epithelium of the pancreatic duct”. This sounds awkward; these lesions do not induce dysplasia but can progress towards different stages of dysplasia to invasive cancer.Author Response
We express our thanks to the reviewer for their time to provide constructive feedback on the article and hope that they find our responses and subsequent changes to the document suitable for the article’s endorsement.
What is somewhat missing is a ‘quality control’ of the included papers. How reliable were the effects on EMP analysed?
We appreciate concerns of the inclusion of references without a “quality control” of the robustness of their findings. Without the direct inclusion of the papers’ authors we find it difficult to comment on the strength of each paper’s findings, and hope that the scientific community aims to validate published findings through independent research. One aim of the table was to supply the reader with the mechanistic findings and the assays employed as a means of assessing the strength of the finding. Certainly, papers that measured EMP by RTqPCR and WB are a very basic starting point when compared to papers that have performed in vitro and in vivo invasion/metastasis/ therapy response assays. We have added the below text to into the table introduction paragraph to highlight the importance of the extension of published findings by the field:
“The proposed mechanisms of candidates and assays used to assess such effects are shown in tables, and may be used to gauge where further support may be warranted by the field to extend and solidify these findings.”
Was EMP the major focus of the studies or only a ‘bystander’ result?
We thank the reviewer, and have updated the table to classify the articles as those having either a “direct” or “Indirect” focus on EMP. Many articles premise their research on candidates that are abundant in their particular cancer models and tissue, whereby a link is discovered to migration/invasion or patient metastases. These papers, and those that solely focussed on EMP as an endpoint were classified as having a “Direct” EMP observation, while those articles where the EMP association was incidental were classified as having “Indirect” EMP observation.
Further, a graphical overview about the most common pathways analyses would be helpful; e.g. many analysed genes were related (directly or indirectly) to TGF-beta signalling.
We appreciate feedback on the readability of the article, and hope that Figures 1 and 2 aid the reader to understand the context of candidates in EMP.
There are several incorrect statements:
We appreciate the technical feedback and hope the below modifications to text in the document are suitable:
e.g. four major oncogenic pathways (actually, one oncogenic, three tumor suppressor);
These studies have identified four major genetic aberrations common to pancreatic tumours
95% inactivating mutations of CDKN2 (actually, mutated only partially and otherwise silenced).
whilst 95% contain either partial or complete inactivating mutations in CDKN2A, contributing to loss of cell cycle regulation
“a panel of 32 consistently mutated genes” is not correct. Probably it is meant “genes in 32 specific pathways”.
As well as these four common driver mutations, genomic sequencing of tumours has identified an additional panel of consistently mutated genes. These genetic mutations implicate functional pathways often dysregulated in cancer, including KRAS, TGFβ, WNT, NOTCH, ROBO/SLT, G1/S, SWI-SNF, and chromatin/ DNA/ RNA modification and repair.
However, we draw the reviewer’s attention to findings from Bailey et al 2016, Nature in reference to specific numbers of significantly mutated genes (22-32) in 10 aggregate pathways depending on computational method of mutational analyses.
“…PanINs, induce dysplasia within the epithelium of the pancreatic duct”.
This sounds awkward; these lesions do not induce dysplasia but can progress towards different stages of dysplasia to invasive cancer.
Such neoplasms, namely PanINs, progress through stages of dysplasia within the ductal epithelium, giving rise to the most common form of PC, pancreatic ductal adenocarcinoma (PDAC)
Reviewer 2 Report
This paper review the role of epithelial mesenchymal plasticity (EMP) in pancreatic cancer, and the preclinical target discovery. However, this theme is not interesting but general, and it is poorly written and not concrete.
What is the critical point compared with other review?
There are too many tables and it is hard to get clearer insight from readers in this field.
It will be better to summarize the figures from tables.
This review in this form is not appropriate for the publication of "Cancers".
Author Response
We hope that the refinements made in response to the specific requests of the other 3 reviewers, all of whom were otherwise quite positive about the importance of our subject and the novelty of our approach, albeit comprehensive, will satisfy the concerns raised by this reviewer. In particular:
Addition of Figures 1 and 2 and graphical overview and schematic of most important mediators of EMP in PC cells and possible link to candidates discussed (Reviewers 1/3,4)
Inclusion of the type of prognostic association and number of samples used to assess effect (reviewer 4)
A comment on the “quality control” of studies included (Reviewer 1), an introduction of promising candidates into the text (Reviewer 4), and a discussion of the potential utility of their combined analysis (Reviewer 4)
Reviewer 3 Report
The review “Targeting Epithelial Mesenchymal Plasticity in Pancreatic Cancer: A Compendium of Preclinical Discovery in a Heterogeneous Disease” is a contribute to the field since it summarizes concisely the process of EMP in the very agressive pancreatic cancer and the observations that have been recently acquired in different in vitro and vivo pre-clinical models. Furthermore, this review presents the evidences in a very-well organized way and summarizes a breath of studies in comprehensive tables that may be very useful for the readership. it also relates to the importance of the pre-clinical research to the adequate patient stratification regarding clinical efficacy.
I recommend publication of the manuscript after the following questions are addressed.
Minor changes:
Line 24: common cause of cancer-related death”s”;
Line 66: that that enables them.. delete one that;
Line 92: define pT stage;
Line 117: replace metastasise by metastasize;
Line 121: levels that are insufficient;
Line 127: consider replace dissection by examination;
Line 129: define PKCY;
Line 157: member associated proteins instead of factors.
The titles of the tables should be more exhaustive.
Although not completely necessary, a Figure regarding the most important mediators of EMP and their main targets in the PC cells could be added to illustrate their importance.
Author Response
We express our thanks to the reviewer for their time to provide constructive feedback on the article and hope that they find our responses and subsequent changes to the document suitable for the article’s endorsement.
Line 24: common cause of cancer-related death”s”;
Corrected
Line 66: that that enables them.. delete one that;
Corrected
Line 92: define pT stage;
Corrected to: ZEB2 overexpression in tumour-stroma associated cells also correlated with pathological assessment of tumour size, and lymph node metastasis
Line 117: replace metastasise by metastasize;
Corrected
Line 121: levels that are insufficient;
Corrected
Line 127: consider replace dissection by examination;
Corrected
Line 129: define PKCY;
Corrected to define in line 96: Such debate has necessitated the use of genetically engineered mouse models (GEMMs) to trace the role of EMP in cancer progression, specifically the pancreatic epithelium conditional Kras/P53 mutant (PKCY) mice
Line 157: member associated proteins instead of factors.
Corrected
The titles of the tables should be more exhaustive.
We appreciate the feedback to include more description of the tables’ contents, and hope that modifcations to titles are more helpful
Although not completely necessary, a Figure regarding the most important mediators of EMP and their main targets in the PC cells could be added to illustrate their importance.
We appreciate feedback on the readability of the article, and hope that Figures 1 and 2 aid the reader to understand the context of candidates in EMP.
Reviewer 4 Report
This is a timely review on an interesting subject.
The background and the cited literature are up-to-date and properly discussed, and this study is in line with the recently renewed interest in targeting EMP as alternative cure strategy in pancreatic cancer. The data, presented with clear and exhaustive tables, are sustained by appropriate background and the conclusions are well interpreted.
A number of reviews have already evaluated the different facets of EMP in pancreatic cancer, including those focused on molecular and epigenetic mechanisms of EMP regulation and metastasis, therapy development and resistance. However, this review focuses on some of the ongoing controversies surrounding in vivo evidence of EMP and the limitations of current approaches, highlighting the need to integrate a greater diversity of published EMP molecular regulators.
There are, however, several comments listed below that the Authors should take into consideration during the revision of their manuscript:
The authors should explain better what is the "prognostic association" reported in the table (number of patients, type of biomarker (mRNA or protein), from tumor tissues or other specimens, etc...) Why not include at least some examples of the potentual candidates in the text? At least one or 2 figures summarizing the molecular mechanisms underlying the activity of these cadnidates in EMP are needed The authors should underline in the discussion how they selected the reported studies and what is the added value compared to other published articles The authors should also be prompted to at least discuss some potential combined analysis of the markers
Author Response
We express our thanks to the reviewer for their time to provide constructive feedback on the article and hope that they find our responses and subsequent changes to the document suitable for the article’s endorsement.
The authors should explain better what is the "prognostic association" reported in the table (number of patients, type of biomarker (mRNA or protein), from tumor tissues or other specimens, etc...)
We thank the reviewer for their feedback to to improve the clinical utility of the article. We hope that updates and corrections in the table listing prognostic association type, and inclusion of number of patients surveyed by mRNA or IHC is suitable.
Why not include at least some examples of the potentual candidates in the text?
Added to text:
Figure 2 illustrates the proposed activity of some of these novel candidates, and how they may positively or negatively regulate discrete EMP signaling pathways. Of note are several candidates that converge to positively regulate EMP migratory phenotypes through FAK/Src and FAK/PI3K signaling, including the 5HT receptor and mucins, as well as EEF2K, USP22, and ZIP4. Their complete mechanisms of action and prevalence in PDAC tissue remain to be elucidated, however their inhibition may curb carcinoma invasion by blocking FAK activation and subsequent EMP modulation. Similarly, candidates participating in stability of EMP signaling and TF activity provide targets to modulate the EMP process specific for carcinoma cells. AURKA kinase has been shown to participate in a positive feedback loop with stabilization and activity of TWIST1, while PEAK1 and NES have been implicated in stabilization YAP/TAZ and SMAD TF activity. The discovery of discrete EMP regulation and development of combinatorial inhibitors may provide the opportunity for more personalized therapeutic approaches to curb metastatic disease.
At least one or 2 figures summarizing the molecular mechanisms underlying the activity of these cadnidates in EMP are needed
We appreciate feedback on the readability of the article, and hope that Figures 1 and 2 aid the reader to understand the context of at least some of the exhaustive list of candidates in EMP.
The authors should underline in the discussion how they selected the reported studies and what is the added value compared to other published articles
Added to text:
The candidates shown were selected by searching Pubmed for the terms ‘pancreatic’ and ‘epithelial’, and articles referencing a novel candidate’s impact on EMP phenotypes were selected for inclusion.
The authors should also be prompted to at least discuss some potential combined analysis of the markers
Added text to discussion:
While candidates have been shown to play roles in aspects of EMP signaling and associated phenotypes, significant support is required for their mechanisms of action to make concrete conclusions about their directive actions in cancer. Our understanding of receptor mediated canonical signaling through PI3K/AKT, MAPK, NFkB and other well studied cell cycle pathways has required decades to tease apart, and the subtleties of EMP programs provides a similar challenge. Open source integrative tools such as Reactome [193], WikiPathways [194], String[195], and Cytoscape [196] provide platforms for researchers to combine such analyses to build upon our current understanding and fill knowledge gaps in the field of cancer biology. In this way, progress may be made to better understand and discover properties that may be modulated in concert to control EMP in cancer.
Round 2
Reviewer 1 Report
The authors have satisfactorily answered some of the questions/concerns of the reviewer. However, the main concern remains that the authors do not report the methodology they used to select the factors and that there is no ‘quality control’ of the included papers. There should be a critical analysis of these papers. They are just reported and displayed.
The novel figures 1 and 2 are confusing. For example (figure 1). TGF-beta is signalling via p-Smad. Then there is an arrow pointing to a box with lncRNA, miRNA, ZEB, Twist, SNAIL. Does it mean TGF-beta is inducing expression of all these factors? Are the same effects mediated via Ras signalling?
Similarly figure 2 suggests that beta-catenin can be activated by WNT signalling or Akt. How is this related to EMP and migration?
Author Response
Reviewer 1 further changes:
The authors have satisfactorily answered some of the questions/concerns of the reviewer. However, the main concern remains that:
the authors do not report the methodology they used to select the factors:
We appreciate the reviewers response to clarify how candidates were chosen for inclusion and hope they find the inclusion of text below suitable:
“The candidates shown were selected by searching Pubmed for the terms ‘pancreatic’ and ‘epithelial’, and articles were manually curated for their investigation into a novel candidate’s role on EMP phenotypes.”
and that there is no ‘quality control’ of the included papers. There should be a critical analysis of these papers. They are just reported and displayed.
We appreciate the reviewers response to clarify their concern regarding QC of papers, and hope they find the inclusion of text below and bolding of factors that showed clinical correlation with lymph node invasion as a gold standard for significance of findings in an EMP context. We hope that the discussion of several factors as a prelude to the table emphasises some of the more significant findings
“The proposed mechanisms of candidates and assays used to assess such effects are shown within tables, and may be used to gauge where further support may be warranted to confirm and extend such findings. Due to the inherent variation in models used, the statistical power granted by IHC for varying sized patient cohorts with accompanying clinical information, and the level of EMP as a primary context, it is difficult to draw direct conclusions regarding pivotal significance within the field and clinical importance from such singular studies. Candidate expression in primary patient material that correlated with lymph-node metastasis are shown in bold within tables, and provide the best surrogate for their role in EMP mediated invasion, and include membrane bound proteins IGFBP2, ITGB4, CEACAM6 [96-98]. The use of IHC to capture dynamic EMP processes may be limited however, as shown in the case of LIN28B, where its expression is both induced by TGFβ and high in PDAC tissue, despite its role to suppress the pro-EMP non-coding RNA LET7a [99,100]. Such studies highlight both the utility and limitations of the links between in vitro assays and clinical material, and emphasise the need for both wider cohorts of patient material for validation and the development of GEMM models to strengthen findings in a standardized manner.”
The novel figures 1 and 2 are confusing. For example (figure 1). TGF-beta is signalling via p-Smad. Then there is an arrow pointing to a box with lncRNA, miRNA, ZEB, Twist, SNAIL. Does it mean TGF-beta is inducing expression of all these factors? Are the same effects mediated via Ras signalling?
We thank the reviewer for their feedback regarding figures. Figure 1 aims to show major pathways that modulate transcriptional control of EMP TFs, lncRNAs and miRNAs rather than induction of specific transcriptional regulators and the subsequent complex feedback loops, for which the readers are referred to reviews covering such topics. We have altered the text and figure caption to better illustrate that these figures are simplified overviews of complex pathways, for which each may deserve a figure in itself, however was beyond the scope of this review to outline the possible roles for novel candidates within these frameworks.
Text referencing figure 1 altered:
EMP is modulated by receptor tyrosine kinases (RTK) ligands, TGFβ, hypoxia via HIF1α signalling, as well as HIPPO, NOTCH, Interleukin and WNT ligands. Their mechanisms and specific impact on downstream EMP targets have been comprehensively reviewed elsewhere, however our understanding of their subtleties and feedback loops is on-going [82,83]. In brief, these complex pathways serve to influence transcriptional programs that co-operate directly and indirectly to control the plasticity that exists between epithelial and mesenchymal phenotypes of carcinoma cells (Figure 1). Of note is the increasing recognition for the role of long non-coding RNAs (LncRNA) and micro-RNAs (miRNA) in EMP regulation.
Figure 1 altered:
ZEB/SNAIL/TWIST changed to EMP transcription factors
Bracket shortened to not include RhoA
B-catenin removed from nucleus
Caption altered: Underlined lines have been added to aid figure description
EMP is regulated by multiple signal transduction pathways that cooperate directly and indirectly to induce a spectrum of EMP phenotypes through transcriptional modulation of EMP transcription factors, long non-coding RNAs (LncRNA), and micro RNAs (miRNAs). These factors then modulate EMP by discrete regulation of epithelial (Red Box) and mesenchymal (Green box) cellular properties, which in turn influence migration and invasion. Transforming growth factor (TGFB) activates the greatest number of these pathways, including direct cytoskeletal regulation by RhoA, aswell as canonical SMAD and non-canonical p38/JNK, MEK/ERK MAPK pathways and PI3K/AKT. Receptor tyrosine kinase (RTK) signalling is induced by binding of growth factor (GF) ligands such as EGF, IGF, FGF, HGF or VEGF and activates RAS/MEK/ERK, PI3K/AKT/NFκB and downstream SRC pathways. WNT signalling also modulates EMP by downstream stabilisation of B-catenin and subsequent nuclear translocation for EMP program activation by TCF/LEF transcription factors. Interleukins (ILs) can also induce EMP programs via STAT3 signalling. Additional mediators of EMP include Hypoxia, Hedgehog, Notch and Integrin signalling (not shown), and highlight the context dependent activation of EMP from micro-environmental cues.
Similarly figure 2 suggests that beta-catenin can be activated by WNT signalling or Akt. How is this related to EMP and migration?
For clarity, figure 2 has been altered for B-catenin arrow to point directly to EMP. Given the complexity and overlap within signal transduction pathways influencing EMP, these pathways have had intermediates removed, except where required, to allow for easier visualisation of role of novel candidates. References to the mechanisms of EMP induction have been provided with the aim of providing a fuller picture of pre and post transcriptional regulation of these processes. B-catenin can be positively regulated by both AKT and Wnt signalling whereby it translocates to the nucleus to associate with TCF/LEF transcription factors and drive EMP. This pathway is included to show the novel association that RAC1 possesses to also activate B-catenin. DKK3 also acts as an antagonist to block Wnt signalling in this context.
Text reference to figure 2 altered:
"Figure 2 illustrates the proposed activity for some of these novel candidates, and how they may positively or negatively regulate discrete EMP signalling pathways. Of note are several candidates that converge to positively regulate EMP migratory phenotypes through FAK/Src and FAK/PI3K signalling, including the 5HT receptor and mucins, as well as EEF2K, USP22, and ZIP4. Their complete mechanisms of action and prevalence in PDAC tissue remain to be elucidated, however their inhibition may curb carcinoma invasion by blocking FAK activation and subsequent EMP modulation. Similarly, candidates participating in stability of EMP signalling and TF activity provide targets to modulate the EMP process specific for carcinoma cells. AURKA kinase has been shown to participate in a positive feedback loop with stabilization and activity of TWIST1, while PEAK1 and NES have been implicated in stabilization YAP/TAZ and SMAD TF activity. The discovery of discrete EMP regulation and development of combinatorial inhibitors may provide the opportunity for more personalized therapeutic approaches to curb metastatic disease."
Reviewer 2 Report
This revised form is now acceptable for publication of "Cancers".
Author Response
We thank the reviewer for their endorsement.
Round 3
Reviewer 1 Report
see previous reviews
Author Response
We thank the reviewer again for their time